# BIAPSS: A Comprehensive Physicochemical Analyzer of Proteins Undergoing Liquid–Liquid Phase Separation

**DOI:** 10.3390/ijms23116204

**Published:** 2022-05-31

**Authors:** Aleksandra E. Badaczewska-Dawid, Vladimir N. Uversky, Davit A. Potoyan

**Affiliations:** 1Department of Chemistry, Iowa State University, Ames, IA 50011, USA; abadacz@iastate.edu; 2Department of Molecular Medicine and USF Health Byrd Alzheimer’s Research Institute, Morsani College of Medicine, University of South Florida, Tampa, FL 33612, USA; 3Bioinformatics and Computational Biology Program, Iowa State University, Ames, IA 50011, USA

**Keywords:** liquid–liquid phase separation, membraneless organelles, intrinsically disordered proteins, proteins with low complexity

## Abstract

The liquid–liquid phase separation (LLPS) of biomolecules is a phenomenon which is nowadays recognized as the driving force for the biogenesis of numerous functional membraneless organelles and cellular bodies. The interplay between the protein primary sequence and phase separation remains poorly understood, despite intensive research. To uncover the sequence-encoded signals of protein capable of undergoing LLPS, we developed a novel web platform named BIAPSS (Bioinformatics Analysis of LLPS Sequences). This web server provides on-the-fly analysis, visualization, and interpretation of the physicochemical and structural features for the superset of curated LLPS proteins.

## 1. Introduction

The spatiotemporal organization of biomolecules and biomolecular interactions is essential for the efficient regulation of cellular biochemistry. The underlying biophysical mechanism for membraneless compartmentalization is liquid–liquid phase separation (LLPS). In the past few years, the LLPS of biomolecules has become a unifying physical mechanism for understanding the principles of intracellular compartmentalization, the formation of membraneless organelles (MLOs), and gene regulation [1,2,3,4,5,6,7,8,9,10,11,12,13,14]. In the LLPS process, the relatively well-mixed solution of biomolecules separates into liquid droplets. The ability of proteins to phase separate appears to be encoded primarily in the peculiarities of their primary sequences, which often contain low-complexity regions and intrinsically disordered regions (IDRs) that are enriched in charged and multivalent interaction centers [6,7,8,10,11,13,14,15,16,17,18,19]. While some general sequence trends have emerged, the quantitative aspects of how amino acid sequences encode and decode phase separation still remain largely unknown [20,21,22]. This is because many different combinations of relevant interactions seem to be contributing to phase separation, without any one being universally necessary [23]. As a consequence (with a few exceptions [24,25,26,27,28,29,30]), mostly case-by-case studies of different sequences are performed, with the broader context of many findings, including their statistical significance, remaining unknown.

Following the statistical trends in PubMed, biological LLPS has been gaining widespread attention in the last two decades. The rapidly growing amount of data from both in vitro and in vivo experiments have systematically narrowed the range of the LLPS-promoting conditions [31]. From these studies, we know that the regulatory mechanisms of phase separation appear strongly context-dependent [31]. The key factors include: the physicochemical state of the protein (e.g., posttranslational modifications), the environmental conditions (e.g., temperature, pH), and the concentration of binding partners (e.g., proteins, nucleic acids, carbohydrates, lipids). Many among the recent hypotheses suggest the prevalence of: (i) electrostatics and π-stacking; or (ii) specific sequence decoration in charge or hydrophobicity; and (iii) the role of short sequential (e.g., GARs (glycine–arginine-rich) [32]) or structural (e.g., LARKS (low-complexity amyloid-like reversible kinked segments) [33]) motifs [15]. However, deciphering the interplay between sequence composition and phase separation turns out to be challenging.

In recent years, several databases have emerged that collect LLPS-related protein sequence data and metadata, with prominent examples being PhaSepDB [25], PhaSePro [26], LLPSDB [27,28], and DrLLPS [29]. These databases collect and annotate partially overlapping sets of phase-separating protein sequences, including data on the experimental conditions and significant annotations. In particular, PhaSePro, LLPSDB, and a subset of PhaSepDB contain manually curated proteins, which are recognized for driving the formation of subcellular compartments.

The accumulation of high-quality datasets is certainly a necessary condition for making progress towards uncovering the driving forces of protein phase separation. However, one needs a biophysically motivated computational infrastructure to be able to harness the data from carefully and manually curated sets of phase-separating proteins for revealing the molecular features that determine protein phase separation. We argue that providing the concise but informative patterns of various features, all together horizontally stacked along the protein sequence, could improve the identification of the significant yet nontrivial correlations that contribute to the multivalent interactions. On the basis of these premises, we have developed a novel web platform named BIAPSS: Bioinformatics Analysis of Liquid–Liquid Phase-Separating Protein Sequences (available at https://biapss.chem.iastate.edu/ and last accessed on 31 May 2022). BIAPSS combines a high-throughput interactive deep sequence analysis with a comprehensive pre-parsed bioinformatics database containing a wide array of physicochemical and evolutionary features that are relevant for low-complexity, disordered, and ordered proteins. This platform provides scientists working in the field of biomolecular condensates with a versatile tool for the rapid and on-the-fly deep statistical analysis of LLPS-driver protein sequences.

## 2. Results and Discussion

### 2.1. Introduction of BIAPSS

Figure 1 represents the features included in the comprehensive BIAPSS analyzer. These features combine sequence composition and biophysical properties. The composition component is represented by: (i) the amino acid content, including frequencies and patterning (i.e., distribution and enriched regions); and (ii) the sequence complexity, which comprises the detection of low-complexity regions, repeats, short motifs, and the on-the-fly calculation of Shannon entropy. The biophysical component covers the physicochemical and structural properties. Specifically, we provide a set of residue-resolution patterns: polarity, hydrophobicity, aromaticity, charge induced interactions and hydrogen bonding. These properties, correlated with the experimentally confirmed LLPS regions, facilitate the identification of the nature and driving forces of interactions. The structural properties aid in filtering out the interactions involved primarily in stabilizing the structure or in identifying regions prone to disorder-to-order transitions.

Previously, such structural switchers were recognized in low-complexity and internally disordered sequences that function via phase separation [34]. Thus, the collected molecular properties incorporate robust sequence-based predictions for the secondary structure, solvent accessibility, intrinsic disorder, and intramolecular contacts. Finally, the evolutionary context derived from the joint outcome of the HMMER-based analysis [35] and Pfam database search [36] highlights the location of functional domains, including those specialized in nucleic acid recognition. The other highly conserved short motifs or individual positions detected through the analysis of multiple sequence alignments may confirm that evolution deliberately preserves phase separation.

The comprehensive approach adopted in BIAPSS (Bioinformatics Analysis of LLPS Sequences) consists of integrating multiple third-party tools and high-performance computing, followed by in-house biostatistical analysis and the extraction of meaningful results (see the Materials and Methods section). The protocol was successfully applied for 501 proteins with experimental evidence of phase separation. Moreover, to the best of our knowledge, the resulting platform represents the broadest database with physicochemically characterized LLPS proteins (see Figure 2). In particular, the pool of entries completely covers the contents of several primary databases of curated LLPS deposits (PhaSePro, PhaSepDB.v1, LLPSDB), which collect annotations and experimental conditions. High interest in the phase-separation phenomenon has already spurred the growth of experimental data repositories. However, the deficiency of the computational infrastructure that targets the integrated biophysical and statistical analysis of phase-separating systems still hampers progress in the field. Therefore, in addition to the open access to the raw yet standardized and well-documented results of our extensive work, we have developed a web-based BIAPSS platform for interactive customized exploration and easy interpretation.

As a user-friendly web server, BIAPSS (https://biapss.chem.iastate.edu/, last accessed on 31 May 2022) is billing itself as a central resource for the systematic and standardized statistical analysis of the biophysical characteristics of the known LLPS sequences.

The web service provides users with:(i)A database of the superset of experimentally evidenced LLPS-driver protein sequences.(ii)A repository of precomputed bioinformatics and statistics data.(iii)Two sets of web applications supporting the interactive analysis and visualization of the physicochemical and biomolecular characteristics of LLPS proteins.

The applications integrate the results from our comprehensive computational approach. The SingleSEQ module includes a residue-resolution biophysical analyzer for interrogating individual protein sequences. The complementary analyses are organized in nine web applications that toggle between a generalized summary view and details specific to a given characteristic. The latter allows users to correlate regions prone to phase separation with an array of physicochemical attributes, structural properties, detected domains, and various sequential or structural motifs. Many characteristics provided by applications in the SingleSEQ pipeline are qualitative and show a profile or pattern of the feature along the amino acid sequence. Examples include distributions of physicochemical characteristics, such as polarity, hydrophobicity, charge, residues forming hydrogen bonds, and pi-stacking. In these cases, the assignment is binary, and the numerical value is the percentage of residues in the sequence that meet the criterion. The second group of characteristics includes structural features predicted based on the amino acid sequences with top-ranked tools. Here, examples include the secondary structure, the solvent accessibility, the tendency to disorder, and low-complexity regions. The visual representation is developed to assign each position along the amino acid sequence a discrete consensus value (e.g., helical or extended, or coil for the secondary structure). The numerical value is the percentage of residues that meet the given criterion (e.g., % of helical). Figure 1 is a concept image, while, in the interactive graphs, there is a label of what the given value refers to. Furthermore, for those interested in in-depth analyses, the individual applications offer an on-the-fly exploration of the results from the original tools, which typically provide the fractional probabilities for each variant of a feature (e.g., p(helical), p(extended), p(coil)) for each position along the protein sequence.

BIAPSS also includes the MultiSEQ module. One of its aims is to obtain insight into the overall characteristics of the sufficient nonredundant set of LLPS-driver protein sequences. The comparison to the benchmarks of various protein groups enables a statistical inference of specific phase-separating affinities. Finally, BIAPSS incorporates an extensive cross-reference section that links all entries to primary LLPS databases and other external resources, thereby serving as a central navigation hub for the phase-separation community. All the data used by BIAPSS are freely available for download as well-formatted files with detailed descriptions, facilitating rapid implementation in user-defined computational protocols. The long-term plan for BIAPSS is for it to serve as a unifying hub for the experimental and computational community. Thus, it provides a comprehensive set of analytic tools, biophysically featured data, and standardized protocols that facilitate the identification of the sequence signals that drive the LLPS, which altogether can support applications for designing new sequences of biomedical interest.

### 2.2. Case Study and Tutorial: Fused in Sarcoma (FUS)

To illustrate the practical utility of BIAPSS, we carefully interpreted the results for fused in sarcoma (FUS) (UniProt ID: P35637), which is a widely used model system to study biological phase separation [37]. We provide below the details of the BIAPSS-based analysis, combined with a handy tutorial on the BIAPSS functionalities.

Fused in sarcoma (FUS) is one of the early discovered biological systems that undergoes self-organization by liquid–liquid phase separation (LLPS) [37]. Since then, the protein has been the subject of extensive experimental and computational research to understand the molecular mechanisms and interactions that drive this phenomenon. FUS can be found in the SingleSEQ module of the BIAPSS service by the UniProt identifier (P35637), the gene (FUS), or by using the “RNA-binding” search key (Last accessed on May 30 2022). The summary page contains a high-quality image of the experimentally confirmed cellular location (left panel in Figure 3). Due to its multifunctionality in RNA processing, FUS is mostly observed in the nucleus [38]. In physiological conditions, the low levels of the protein are distributed in the cytoplasm [39], where FUS transports and manages RNA through the dynamic liquid-like subcellular compartments, such as ribonucleoprotein or stress granules [40]. However, the cytoplasmic concentration of FUS significantly increases when noxious mutations lead to aggregation [41].

This progressively aberrant process is manifested by neurodegenerative diseases in humans [41]. Although plenty of accumulated evidence points to the influence of distinct factors on the cellular behavior of FUS, its primary sequence still holds many cues. To frame the physicochemical properties of full-length FUS, we used the analytical approach offered by the SingleSEQ module of BIAPSS.

The average metrics, available in the *Summary* of the SingleSEQ module, indicate that the 526-residue-long sequence of FUS contains over 80% disorder and only 8% order. The solvent-accessibility predictions show the same aspect ratio between exposure and burial. The contents of aromatic, hydrophobic, polar, and charged residues are 10%, 42%, 40%, and 17%, respectively, with a slight excess of positive charge. Such a rough overview described by a set of averages gives some general insight into the protein properties, but it conceals some local distributions that are important for the identification of the preferential interactions.

Therefore, we conduct a detailed analysis of the composition and complexity of the FUS sequence, and we present the resulting patterns in Figure 4. Compared to any reference set of proteins (use *Composition and Complexity* app), this one is extremely enriched in glycine, which makes up nearly 1/3 of the full sequence. Another 20% of the amino acid content consists primarily of serine and glutamine. Although the dominant content of these three amino acids suggests the generally low complexity of the sequence, their distribution along the sequence is strongly heterogeneous. Indeed, the calculated low information content of the sequence is mainly localized around protein terminals and clearly corresponds to three fragments with high glycine concentrations (LCR2: residues 164–267; LCR3: residues 370–420; LCR4: residues 454–507). These regions also exclusively accumulate total arginines, which, together with glycine, form a series of RGG repeating motifs that are known to bind RNA specifically [32]. Both serine and glutamine are mostly localized at the N-terminus, being more clearly clustered within LCR1 (1–163). LCR1 additionally gathers 24/35 available tyrosines, and, thus, it has visibly distinct enrichment (SQYG) that is known to occur in prion-like domains (PLD) [43]. By using the *Domains*, *Motifs*, *Repeats* application, we also found that the remaining compositionally more complex regions of the C-terminus (I287-L365 and R422-D453) match the PF00076 and PF00641 Pfam domains (i.e., the RNA recognition motif (RRM) and RNA-binding zinc finger (ZnF), respectively). The robust predictions (for details, see Methods) unanimously show that RRM is a well-folded FUS domain, while the other fragments remain disordered.

The seed MSAs prepared for FUS within the *Sequence Conservation* application further confirm that both domains are evolutionarily conserved members of Pfam families: RRM_1 and zf-RanBP, respectively (see bottom rows in Figure 4). The visual inspection of the amino acid content and the distribution of FUS allows us to identify and isolate specific regions in the protein (Figure 5). Furthermore, we have performed a physicochemical featurization of these segments, using the *Chemical Properties Patterns* app, which reveals the preferred interactions when coupled with biomolecular conditionals that are known from experiments. The recent experimental reports show that the isolated prion-like domain (PLD) (residues 1–214, or even residues 1–163) can undergo self-organization, forming liquid droplets when kept at high protein levels or high salt concentrations [44,45]. This N-terminal fragment is enriched in amino acids, whose side chains are multivalent, as shown in Figure 5. Thus, the dense pattern of polarity comes from the enrichment in S, Q, Y, where Y, Q, and G also provide π-electron centers for π-π-stacking. Most of them are also able to be both donors and acceptors of side-chain protons for hydrogen bonding (HB). In line with this, the intermolecular-interaction profiles derived from simulations of the 120–163 region indicate the most frequent contacts between QQ > QY > YY > SY and other pairs of enriched amino acids [46]. All of these observations suggest that the homotypic phase separation of wild-type PLD monomers is driven by balanced contributions from hydrogen bonding and π-stacking. Indeed, several mutagenesis studies show that Y→A substitution disrupts phase separation by the removal of both components of the interaction, while Y→F mutants are significantly more aggregation-prone, due to the strengthening of the binding via tighter hydrophobic F-π-stacking at the cost of losing HB contributions of polar tyrosine [44,46]. It is also worth noting that the PLD region is completely deficient of positive charge, with a minor net charge per residue of −0.01 (M1-S165: −0.012; and M1-G212: −0.024), which places it within the weak polyelectrolyte region on the CIDER diagram (right panel in Figure 3) [42]. However, an excess of serine and threonine in this region provides an ability to introduce a strongly negative charge through multiple phosphorylations. After phosphorylation, the dominant force becomes electrostatic repulsion, which is known to disrupt both phase separation and aggregation [37]. The central region of the PLD (residues 39–95) was proposed as the core of aberrant fibrils, which, in solid-state form, structured cross-𝛽-sheets [37]. The same structural properties have not been unambiguously confirmed in the condensed phase of liquid–liquid mixing. Undoubtedly, however, our algorithms detected, along this region, structural motifs known as low-complexity amyloid-like reversible folded segments (LARKS) [33]. In our analysis, the most effective predictors of structural properties showed, for these motifs, some tendency towards an extended secondary structure and a slightly increased probability of burial (bottom panel in Figure 5; use *Secondary Structure*, *Solvent Accessibility* and *Structural Disorder* applications of the BIAPSS SingleSEQ module). Interestingly, the prediction in eight-letter notation detected a turn or bend within each of the structural motifs, which explains their flexible nature. These findings, together with the ambiguous experimental results, may suggest some variations in the structural state in the PLD core, and specifically the disorder-to-order transition driven by biomolecular conditionals.

The remaining part of the FUS sequence, referred to as the C-terminus, contains two well-known domains (RRM and ZnF) and three glycine–arginine-rich regions (GARs), which are detected using the *Domains, Motifs, Repeats* application. All components are significant players in binding RNA. Zinc finger supports only the recognition of the specific GGU motif, while the RRM domain and RGG repeats are universal towards a variety of RNAs [47]. Both folded domains of FUS are much less polar than the PLD, as seen from the BIAPSS-based physicochemical features in Figure 5. They also have a lower content of side chains that are able to engage in π-stacking or hydrogen bonding. However, the charged residues are pretty abundant in the composition of RRM and ZnF, which explains the functional role of electrostatic interactions towards the binding of nucleic acids or stabilizing folds via salt bridges [48,49].

All three GARs are the least polar regions of the protein (see Figure 5; use *Chemical Properties Patterns* app of BIAPSS SingleSEQ module). The dense patterning of hydrophobicity arises from glycine excess. The rich π-electron-containing systems, other than aromatic side chains, originate mainly from the abundance of the arginine’s guanidino group. Arginine is also a source of excess positive charge at the C-terminus. The experimental studies consistently confirm that the isolated C-terminus does not undergo phase separation [44]. However, liquid–liquid droplets rapidly occur when mixed with N-terminal monomers [44]. Moreover, the LLPS of full-length wild-type FUS is more robust than the heterotypic mixing of the N- and C-terminals and the homotypic self-assembling of N-terminal monomers [44]. This suggests the higher priority of cation-π (R-Y) stacking over π-π (Y-Y) stacking, while both are reinforced by hydrogen bonds. Another experimental study showed that R→K mutants, who no longer have the ability of π-π-stacking but retain charge, can still undergo phase separation. In turn, R→A substitutions prevent phase separation because they lose the π-system, cation, and ability of side-chain hydrogen bonding. Interestingly, the recent report indicates that stacking interactions, including cation-π (e.g., RY, KF) and especially π-π (e.g., YY and RY, and even RQ), are most robust over a wide range of salt concentrations [45]. The hydrophobic contribution from π-electron-containing systems becomes the main force that strengthens the contact in high salt. In these conditions, the screening of usually dominant electrostatic contributions is significant. Surprisingly, changing the partitioning of the different forces makes the interaction of the two positively charged arginines attractive under these conditions [45]. The set of diverse chemical groups in arginine is a unique feature among the other amino acid side chains. With its high reactivity, the need for precise regulation comes, and so arginine can be tuned to a preferred state by posttranslational methylation.

Thus, under physiological conditions, FUS is highly methylated [44]. This limits self-assembly via interactions with tyrosine and promotes a functional role of intermolecular interactions with other proteins and nucleic acids. Therefore, phase separation and the gelation of FUS can increase by the hypomethylation of arginines within RGG-rich regions or the insertion of additional ones into the C-terminus [44]. All of these findings come together to demonstrate the significant role of the arginine side chain in phase separation. Tyrosine and glutamine are similarly relevant.

### 2.3. FUS, LLPS Regulated in the Context-Dependent Tuning of Preferred Forces

Note that the original results generated for FUS (identifier: P35637) using the BIAPSS web platform are shown in Appendix A. The findings described in the previous section are briefly summarized below. FUS is a predominantly disordered protein (80%), composed of two functional regions: the prion-like domain (residues 1–212) and the RNA-binding C-terminal fragment (residues 213–526). Surprisingly, although the sequence is more than 65% composed of only five amino acids (G >> S >Q > R~Y), their distribution is highly variable. In particular, low sequence complexity occurs mainly in three structurally flexible glycine–arginine-rich regions. They provide hydrophobicity, π-electron-containing centers, and a positive charge of the C-terminus. In contrast, the enrichment of the N-terminus in serine, glutamine, and tyrosine makes it strongly polar but negatively charged (due to a few aspartic acids), with numerous aromatic centers and side chains capable of hydrogen bonding. These very different physicochemical properties of the protein terminals are sensitive to environmental changes and allow for the context-dependent regulation of the protein’s cellular behavior. In particular, arginine can be tuned to a preferred state by posttranslational methylation [44], while serine/threonine phosphorylation introduces a highly negative charge [37]. Both modifications prevent the formation of aberrant self-assembly. In the first case, the methyl groups hinder cation-π stacking between arginine and tyrosine, limiting the phase separation driven by contacts between the N- and C-terminals. In the second case, phosphorylation introduces strong electrostatic repulsion between the N-termini. It inhibits homotypic phase separation, driven by the π-stacking of aromatic tyrosines. However, both mechanisms have no effect under high salinity due to the significant screening of electrostatics. In such conditions, even interactions of arginines become attractive due to hydrophobic contributions and π-π stacking [45]. Therefore, these observations demonstrate that the peculiar physicochemical properties of amino acid residues play a significant role in phase separation. The multifunctional chemical groups of amino acids make them reactive and multivalent. These features aid in the context-dependent tuning between preferred modes of interactions. They can work synergistically or alternatively, and their regulation depends on the environmental conditions, the state of posttranslational modifications, and the presence of binding partners.

## 3. Materials and Methods

### 3.1. Sequence Complexity and Physicochemical Decoration

#### 3.1.1. Sequence Complexity

Low-complexity regions (LCRs) in proteins are compositionally biased fragments of sequences that often have low amino acid diversity and repeats of short motifs of the sequential or structural kinds. Many reports point to their functional or regulatory roles, frequently also associated with subcellular phase separation [19]. The LCRs of LLPS proteins have been detected by using several state-of-the-art tools, such as SIMPLE [50], CAST [51], fLPS [52], and SEG [53]. The original hits were parsed by in-house algorithms to merge overlapping regions enriched in different amino acids, and only the integrated and unified results have been kept.

Shannon Entropy describes the information content held in data and it is a frequently used measure of protein-sequence complexity. We implemented a module for the on-the-fly calculation of it within BIAPSS services. The typical window length for compositional effects is between 5 and 20. The results can be displayed in:

Residue-resolution mode (residue option; smoother output):
$$S_{i}=\frac{1}{N}\overset{N}{\underset{j=1}{\sum }}S_{\left(j,N\right)}$$
where the Shannon entropy (*S*_(*i*)_) at sequence position *i* is a sum of entropies at all windows containing this position, normalized by the window length (*N*);Window-resolution mode (block option):
$$S_{\left(j,N\right)}=-\overset{AA=20}{\underset{aa=1}{\sum }}f_{aa}log_{2}\left(f_{aa}\right)$$
where the Shannon entropy (*S*_(*j*,*N*)_) at the *j*-th sequence window of the length (*N*) is summed over the fractions (*f_aa_*) of 20 biogenic amino acids. The value is assigned to the center position within the window. The *S*_(*j*,*N*)_ ranges from 0 (where only one residue is present within the sequence window) to log_2_(*N*) (all positions are different). Therefore, the lower the Shannon entropy, the less complex the sequence is.

#### 3.1.2. Physicochemical Decoration

To examine the physicochemical properties of LLPS-driving proteins, we identified, along each sequence, the patterns of polarity (Ser, Thr, Tyr, Gln, Asn, Cys, Met), hydrophobicity (Gly, Ala, Val, Ile, Leu, Pro, Phe), and detected π-stacking centers (Arg, Asn, Asp, Gln, Glu, Gly (note that, due to the lack of the side chain, glycine can stack via π-electrons from a peptide bond and hydrogen bonding via backbone carbonyl or amide), including those within aromatic rings (Phe, Tyr, Trp, His). We also provided the charge-distribution split between positively (His, Lys, Arg) and negatively (Glu, Asp) charged residues. For each feature, both the arrangement along the sequence and the fraction of residues are provided.

#### 3.1.3. Electrostatics

It is well established that the electrostatic interactions often affect the solubility and stabilize the binding interface in the liquid–liquid demixing of biomolecules. The recently proposed charge-decoration parameters emerged as a measure of charge distribution along the protein sequence. In addition to the overall charge content, these descriptors are seen as important factors that shape the protein conformations, especially within low-complexity regions [54]. Following these discoveries, we calculated and compared the charge-decoration parameters; namely:SCD (sequence charge decoration) is implemented following the formulation by Sawle and Ghosh [55];OCS (overall charge symmetry) is implemented following the formulation by Das and Pappu [56];FCR (fraction of charged residues) is defined as a sum of the fractions of positive and negative charges.

### 3.2. HMMER-Based Sequence Conservation and Functional-Domain Detection

The multiple sequence alignment (MSA) and consensus profile were prepared using an efficient HMMER method (*phmmer* + *hmmalign* and *hmmbuild*, respectively), which employs a probabilistic hidden Markov model (HMM) [35], and are significantly more accurate compared to BLAST-based searches. Because some of the LLPS sequences are highly unique (detection of the remote homologs is needed), and because the MSA is reliable if at least several dozen homologous sequences are available, we used sequences selected from various UniProt subsets. Specifically, SwissProt, UniRef50, and UniRef90 differ in the size and increasing sequence identity of entries [57,58,59]. To identify sequence regions with significant evolutionary conservation, we derived three additional MSA-based parameters: strength, diversity, and character. The MSA strength of the sequence conservation informs on how much the specific position is held by evolution. This measure normalizes results from the *hmmlogo* tool to a discrete range from 0 (poorly conserved) to 5 (highly conserved). The *hmmlogo* computes letter heights along the sequence, depending on the information content of the position. The MSA diversity defines the number of different amino acids detected at a given position in the MSA, and is provided in discrete scale from 0 (highly conserved) to 5 (poorly conserved) (0—one, 1—two, 2—three, 3—four, 4—five or six, 5–7 and more amino acids at the aligned position). The MSA character describes the chemical nature of the most common amino acids at a given position in the multiple sequence alignment. We distinguished the following attributes: polar, charge, aromatic, another π-system, hydrophobic, and other (G or P).

Some LLPS proteins are composed of one or more well-known domains. The identification of these functional regions alongside regions of low complexity or disorder can provide additional insights into the regulatory role of phase separation. Therefore, we have performed a Pfam search for all LLPS proteins, reporting the detected domains and incorporating the original Pfam seed-MSAs for corresponding regions of LLPS sequence (instead of full-length ones) to derive more reliable evolutionary conservation descriptors.

### 3.3. Short Sequential and Structural Motifs Specific for LLPS Sequences

Short linear motifs (SLiMs) are short fragments along the sequence, often situated in the intrinsically disordered regions, generally showing high structural flexibility and evolutionary conservation. We systematically detected various short sequential and structural motifs. The implemented algorithms used the list of grouped motifs’ instances, defined by regular expressions, as the keys to search protein sequences prone to phase separation. Among motifs known from the literature as relevant for phase behavior, our analysis includes short structural stretches of protein sequence, such as LARKS [33] and steric zippers [60]; glycine–arginine-rich regions (GARs) [32]; and new sequential repetitive n-mers.

### 3.4. Structural Properties Derived from Sequence-Based Predictions

Bearing in mind the predictive nature of sequence-based methods and, hence, their limited accuracy, comparing several of them and choosing the final consensus has proven to be successful in many approaches. In our study, we comprised predictions from at least three to six widely used tools for each biomolecular characteristic. While almost every method is available as a web server, due to the size and complexity of our analyses, we employed standalone versions. The raw data derived from these standalone tools during the high-performance computing was initially parsed, filtered, and simplified to a uniform CSV format, and deposited in our online repository at https://biapss.chem.iastate.edu/download.html, accessed on 1 April 2022.

#### 3.4.1. Secondary Structure

Protein secondary structure is a regular three-dimensional organization of local fragments along a polypeptide chain. The two most common secondary structural elements are alpha helices and beta sheets. Most of the predictors provide the secondary-structure assignment in 3-letter notation (ss3): H—helix, E—strand, C—coil, while the advanced ones (RaptorX and PORTER-5) also deliver more detailed 8-letter notation (ss8): H—α-helix; G—3_10_-helix; I—π-helix; E—β-strand; B—β-bridge; T—HB-turn; S—bend; C–loop. In our benchmark study, we employed five well-established secondary-structure predictors: PSIPRED [61], RaptorX-SS8 [62], PORTER-5 [63], SPIDER-3-Single [64], and FELLS [65].

#### 3.4.2. Solvent Accessibility

Solvent accessibility gives some insight into protein structural flexibility, indicating the exposed patches on the protein surface available for interactions with the solvent molecules. Some surface sites have high evolutionary conservation, which is suggestive of functional or structural importance. Since not many structures of phase-separating IDPs are known, the robust prediction of solvent accessibility can help to identify flexible regions prone to conformational changes upon binding. The assignment of solvent accessibility is usually provided in the 3-letter code: B—buried, E—exposed, M—medium. In our benchmark study, we employed three well-established solvent-accessibility predictors: RaptorX-Property [66], PaleAle 5.0 [67], SPOT-1D [68].

#### 3.4.3. Structural Disorder

The sequence-based predictions indicate regions of increased structural flexibility, usually estimating the disorder probability at a given position in the sequence. Detecting highly flexible regions may support the identification of short sequence stretches of multivalent interactions that can be relevant to phase separation. In our benchmark study, we employed seven well-established predictors of structural disorder: RaptorX-Property [66], IUPred2A [69], SPOT-Disorder [70], DISOPRED (v2 and v3) [71], and PONDR^®^ (FIT, VLXT, VSL2) [72]. Most of these methods return the probability of disorder for each position in the sequence. Usually, the residue is considered as ordered when the score is below 0.5. The protein-binding regions in disordered fragments were estimated using the ANCHOR method [73].

#### 3.4.4. Contact Map

Contact-map application provides a more reduced representation of a protein structure using a binary two-dimensional matrix of distances between all possible amino acid-residue pairs. The commonly used definition assumes the threshold 6–10 Å as the distance between the pair of two Cα or Cβ atoms being in contact. The contact number of protein residues limits the number of possible protein conformations and helps encode a three-dimensional structure. In our benchmark study, we employed three state-of-the-art predictors of intramolecular contacts: RaptorX-Contact [74], ResPRE [75], SPOT-Contact [68,76].

### 3.5. Data Availability

The UniProt IDs of LLPS sequences were collected as a joint superset of deposits from primary LLPS databases (i.e., PhaSePro (https://phasepro.elte.hu/, accessed on 1 April 2022), PhaSepDB.v1 (http://db.phasep.pro/, accessed on 1 April 2022), LLPSDB (http://bio-comp.org.cn/llpsdb/, accessed on 1 April 2022)). Then, protein sequences were taken from the UniProt database, available at https://www.uniprot.org/, accessed on 1 April 2022. The cellular location of the protein was derived via web scraping of primary LLPS databases, UniProt and COMPARTMENTS (https://compartments.jensenlab.org/, accessed on 1 April 2022). The following resources were reviewed for the corresponding entries of experimental or predicted three-dimensional structures: PDBe (https://pdbe-kb.org/, accessed on 1 April 2022), Swiss-Model Repository (https://swissmodel.expasy.org/repository/, accessed on 1 April 2022), ModBase (http://salilab.org/modbase-cgi/, accessed on 1 April 2022), and AlphaFold DB (https://alphafold.ebi.ac.uk/, accessed on 1 April 2022).

The results of our comprehensive analysis, performed on 501 proteins, are available at https://biapss.chem.iastate.edu/download.html, accessed on 1 April 2022. For each file, the details of its content and methods used are comprehensively described. These files are used directly as the input for the web applications of the SingleSEQ and MultiSEQ modules in the BIAPSS platform. The results of the analysis can be explored interactively online, saved as high-quality PNG images, and used directly as figures in the publication.

### 3.6. Code Availability

#### 3.6.1. Phase-Separation Predictors:

1. PSPredictor, web server version available at http://www.pkumdl.cn:8000/PSPredictor, accessed on 1 April 2022.

#### 3.6.2. Low-Complexity-Region (LCR) Predictors (Sequence-Based):

1. SEG, standalone (1999), available at https://ftp.ncbi.nih.gov/pub/seg/seg/, accessed on 1 April 2022;

2. fLPS, standalone Sep 2017, available at https://github.com/pmharrison/flps, accessed on 1 April 2022;

3. SIMPLE, standalone V6-6.1, available at https://github.com/john-hancock/SIMPLE-V6, accessed on 1 April 2022;

4. CAST2, web server version available at http://structure.biol.ucy.ac.cy/CAST2/index.html, accessed on 1 April 2022.

#### 3.6.3. Multiple Sequence Alignment/Build a Profile/Conservation Logo:

1. HMMER (phmmer, hmmalign, hmmbuild, hmmlogo), standalone 3.3, available at http://hmmer.org/download.html, accessed on 1 April 2022;

2. Pfam, the database search was used to detect functional domains, available at http://pfam.xfam.org/, accessed on 1 April 2022.

#### 3.6.4. Secondary-Structure Prediction (Sequence-Based):

1. PSIPRED, standalone 4.02, available at https://github.com/psipred/psipred, accessed on 1 April 2022;

2. RAPTOR-X, standalone Version ID: Rev: 37223, available upon request at http://raptorx.uchicago.edu/download/, accessed on 1 April 2022;

3. PORTER, standalone v5, available at https://github.com/mircare/Porter5/, accessed on 1 April 2022;

4. SPIDER, standalone v3, available upon request at https://sparks-lab.org/downloads/, accessed on 1 April 2022;

5. FESS, standalone 2.0 (November 2016), available upon request at http://old.protein.bio.unipd.it/download/, accessed on 1 April 2022.

#### 3.6.5. Solvent-Accessibility Prediction (Sequence-Based):

1. RAPTOR-X property, standalone v1.01 (October 2018), available upon request at http://raptorx.uchicago.edu/StructurePropertyPred/predict/, accessed on 1 April 2022;

2. PaleAle, standalone 5.0 (December 2019), available at https://github.com/mircare/Brewery, accessed on 1 April 2022;

3. SPOT-1D, standalone (July 2019), available upon request at https://servers.sparks-lab.org/downloads/, accessed on 1 April 2022.

#### 3.6.6. Structural-Disorder Prediction (Sequence-Based):

1. RAPTOR-X property, standalone v1.01 (October 2018), available upon request at http://raptorx.uchicago.edu/StructurePropertyPred/predict/, accessed on 1 April 2022;

2. UPred2A, standalone (November 2019), available upon request at https://iupred2a.elte.hu/download_new, accessed on 1 April 2022;

3. DISOPRED, standalone v2 and v3.1, available at https://github.com/psipred/disopred, accessed on 1 April 2022;

4. SPOT-Disorder2, standalone (February 2019), available upon request at https://sparks-lab.org/downloads/, accessed on 1 April 2022;

5. VSL2, standalone (November 2019), downloaded from http://www.dabi.temple.edu/disprot/download/VSL2.tar.gz (not available now), accessed on 1 April 2022;

6. PONDR-FIT, web server available at http://original.disprot.org/pondr-fit.php, accessed on 1 April 2022;

7. PONDR-VLXT, web server available at http://www.pondr.com/, accessed on 1 April 2022.

#### 3.6.7. Contact-Map Prediction (Sequence-Based):

1. RAPTOR-X Contact, web server available at http://raptorx.uchicago.edu/ContactMap/, accessed on 1 April 2022;

2. ResPRE, standalone (November 2019), available at https://zhanglab.ccmb.med.umich.edu/ResPRE/download/ResPRE.zip, accessed on 1 April 2022;

3. SPOT-Contact, standalone v3 (June 2007), available upon request at https://sparks-lab.org/downloads/, accessed on 1 April 2022.

The raw data collected from the third-party software were parsed utilizing custom python/bash algorithms to provide the unified format and derive the consensus properties. The output files are available at https://biapss.chem.iastate.edu/download.html, accessed on 1 April 2022.

## 4. Conclusions

In conclusion, many proteins undergo liquid–liquid phase separation (LLPS), which drives the biogenesis of various membraneless organelles. The interplay between the protein sequence and the LLPS potential is poorly understood. The BIAPSS web platform, which provides the means for the analysis, visualization, and interpretation of data for LLPS proteins, is designed to uncover the sequence-encoded signals of LLPS proteins. This BIAPSS platform stands out as an efficient and user-friendly visualization framework that facilitates the integration and comparison of the physicochemical and structural features of the vast majority of known phase-separating proteins. With the rapid growth of experimental data on a single-case basis, we expect the increased need for computational infrastructure that consolidates some generalized insights. Hence, we have also developed a feature-rich module for analyzing multiple protein sequences. The interactive interface, with content-rich labels and tooltips, makes data exploration and interpretation easy. Both the web applications and raw datasets are broadly accessible on multiple operating systems and popular browsers. The presented case study of FUS shows that the BIAPSS-inferred biophysical regularities accurately identify regions prone to phase separation and facilitate the design of precise sequence modifications for various applications.

While the current version of BIAPSS enables the convenient and insightful analysis of large nonredundant and high-quality LLPS protein supersets, there are several directions through which our platform could expand to discover unknown LLPS proteins. These include analyzing the physicochemical and structural properties for customized protein sequences and introducing several LLPS indicators trained with machine learning to reveal coupled effects. The new functionality will find applications for flexible sequence redesign to introduce or modulate phase separation. Among many beneficial uses, it could tailor the properties of modern biomaterials or open up new directions in the development of medical therapies.

## Figures and Tables

**Figure 1 ijms-23-06204-f001:**
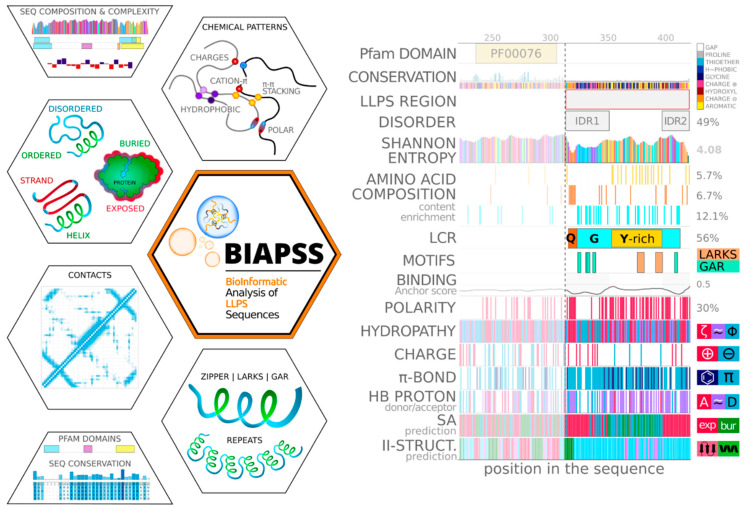
The comprehensive BIAPSS analyzer incorporates the compositional, evolutionary, physicochemical, and structural properties of LLPS proteins. All characteristics can be easily compared and correlated on the horizontally stacked multirow graphs. The interactive exploration helps to filter out sequence signals relevant for phase separation.

**Figure 2 ijms-23-06204-f002:**
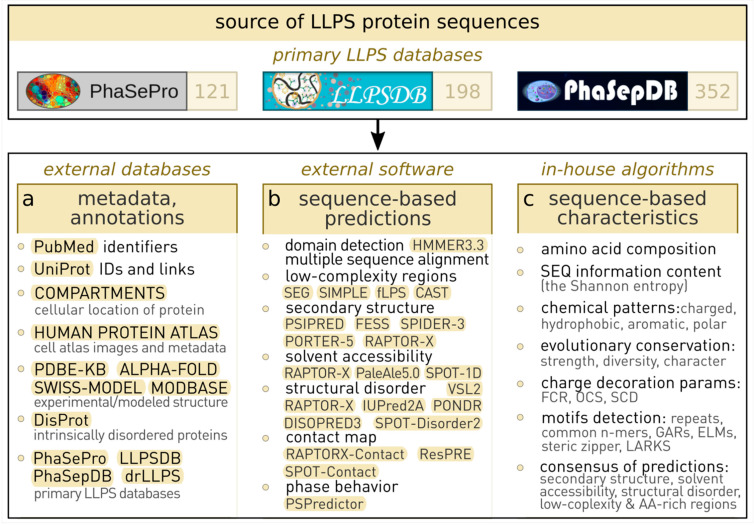
The BIAPSS repository collects the largest dataset of known LLPS proteins that have been identified from carefully curated primary LLPS databases. The computational framework starts from protein sequences downloaded from the UniProt database. The approach builds on three complementary components: (**a**) integration of metadata, annotation, and cross-links from the external databases; (**b**) comprehensive sequence-based bioinformatic analysis of evolutionary and biomolecular properties using state-of-the-art third-party software; and (**c**) meticulous physicochemical and compositional analysis and robust data integration using the in-house algorithms. The BIAPSS interactive web applications enable exploration through the distilled essence of the crafted characteristics.

**Figure 3 ijms-23-06204-f003:**
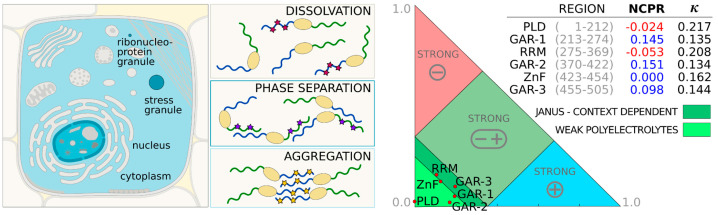
The **left** panel shows the cellular location of FUS (image source: BIAPSS). The protein is predominantly located in the nucleus. The physiological low levels of FUS found in the cytoplasm typically self-organize to membraneless compartments, such as stress granule or ribonucleoprotein granule. The aberrant disease-related aggregates are mostly localized in the cytoplasm. The methylation (purple stars) of C-terminal arginines (green tail) in the wild-type FUS strongly promotes phase separation and gelation. The phosphorylation (magenta stars) of serine and threonine in the N-terminus (blue tail) dissolve liquid-liquid droplets. The tyrosine-to-phenylalanine mutants (yellow stars) in the N-terminus and hypomethylation of arginines in the C-terminus increase aggregation. The **right** panel shows classification of intrinsically disordered ensemble regions (CIDER) [42] for the FUS sequence split into functional segments.

**Figure 4 ijms-23-06204-f004:**
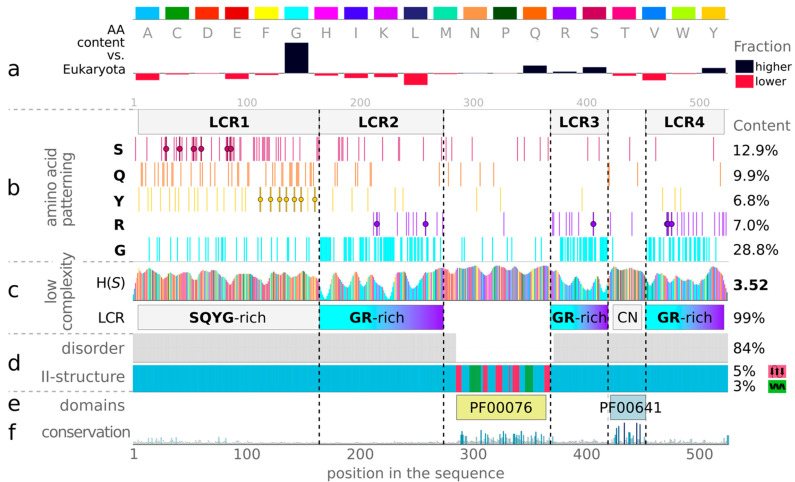
Sequence composition and complexity of FUS. The upper panel (**a**) shows the amino acid (AA) content of the query sequence compared to the Eukaryota dataset (black indicates higher and red the lower content than the reference). The bottom panel shows, the following information for a specific query sequence: (**b**) the patterning of enriched amino acids (S (magenta), Q (orange), Y (yellow), R (purple), G (cyan)); (**c**) low-complexity measures (a color scale corresponding to each amino acid) provided as regions of particular AA enrichments (LCR row), and the sequence information content (H(S) row, Shannon entropy); (**d**) consensus of predicted disorder regions (gray) and secondary-structure assignment (helix in green, strand in magenta, coil in light blue); (**e**) detected Pfam domains; and (**f**) evolutionary conservation derived from the multiple sequence alignment against UniRef50 (blue shades). The amino acid patterning section contains points corresponding to the locations of the most relevant serine phosphorylation sites (residues 30, 42, 54, 61, 84, 87) [37], arginine methylations (residues 216, 259, 407, 472, 473, 476) [44], and tyrosine mutations (residues 113, 122, 130, 136, 143, 149, 161) [44].

**Figure 5 ijms-23-06204-f005:**
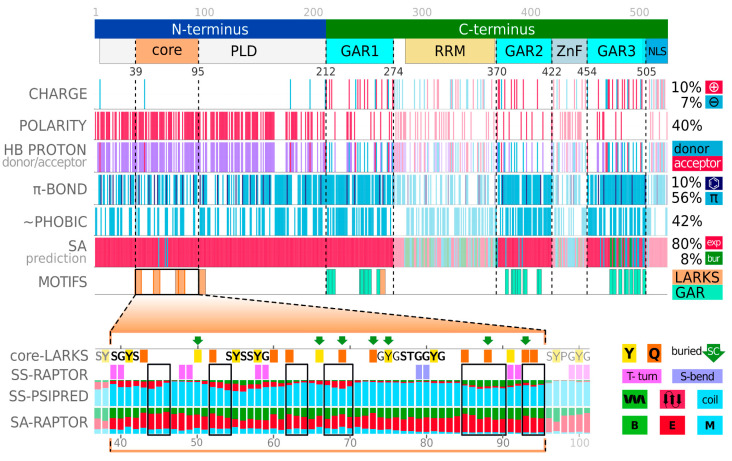
Physicochemical and structural properties of FUS. The various characteristics are shown along the protein-sequence split on the N-terminus (blue) and C-terminus (green). The full-length N-terminus corresponds to a highly polar prion-like domain (PLD in gray). The PLD contains a core region (residues 39–95 in orange), in which multiple LARKS motifs were detected (orange bars in the MOTIFS row) and evidenced to form fibrils in a solid state. The C-terminus contains two well-folded domains detected by Pfam search (RRM in yellow, ZnF in gray), and three glycine–arginine-rich (GAR in cyan) regions. All components of the C-terminus are known to have a functional role in RNA binding. The following rows show the physicochemical patterning along the full-length FUS sequence, including charge (positive (magenta), negative (blue)), polarity, donor (blue)/acceptor (magenta)/both (purple) of side-chain proton for hydrogen bonding, π-electron-containing systems (blue), with separation of aromatic ones (dark blue), hydrophobicity (blue), predicted solvent accessibility (SA) in 3-letter notation (exposed (red), buried (green), medium (blue)). The zoom of the PLD core is shown at the bottom panel, where the SS and SA rows contain the predicted probabilities of secondary structure (helix (green), strand (red), coil (blue)) and solvent accessibility. The green arrows indicate the side chains buried in the fibril core, while the black frames highlight segments that form strands of a cross-β motif [37].

## Data Availability

The data presented in this study are available in this article and in Appendix A.

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
