# Peer review of "BIAPSS: A Comprehensive Physicochemical Analyzer of Proteins Undergoing Liquid–Liquid Phase Separation"

_ijms, 2022, doi:10.3390/ijms23116204_

Round 1

Reviewer 1 Report

The manuscript entitled “BIAPSS: A comprehensive physiochemical analyzer of proteins undergoing liquid-liquid phase separation” is sound and of high interest for the community, meeting the growing and interdisciplinary research community performing research on the topic of liquid liquid phase separation and formation of liquid dense phases.

The manuscript is well prepared and I suggest the manuscript can be published.

However, the authors may comment, as the article is now for more than 1 year in BioRxiv, why they send at this time to IJMS ?

Further Comments:

I suggest the authors use and apply not only FUS, they should include one or two more proteins, and prepare a compact comparative summary of the results obtained by the new sever.

They authors may highlight what the server requires as imput first, UNIProt ID ?

Any option to introduce a amino acid sequence ?

And data obtained are accombined with a reliability factors, or score factors ?

For example Figure 1, left site, data shown in %, have also standard deviations ?

Author Response

Comments for the Authors

The manuscript entitled “BIAPSS: A comprehensive physiochemical analyzer of proteins undergoing liquid-liquid phase separation” is sound and of high interest for the community, meeting the growing and interdisciplinary research community performing research on the topic of liquid liquid phase separation and formation of liquid dense phases. 

The manuscript is well prepared and I suggest the manuscript can be published.

 Authors’ reply: Thank you for this comment.

However, the authors may comment, as the article is now for more than 1 year in BioRxiv, why they send at this time to IJMS ?

 Authors’ reply: We posted an early draft in the bioRxiv preprint system after making the first release of the BIAPSS server publicly available. It was a concise outline of the functionality we were developing and the motivation behind the project. We have also included supplementary materials providing a step-by-step tutorial (now available online in the documentation) for all web applications to allow users to test the server at an early stage of the development. That has resulted in many improvements that we have added to the web platform over the past year. It took some time to work out the details of such a versatile tool. In this manuscript, we present a completed and tested platform showing, by an example, what its capabilities are. We believe that the broad audience of IJMS is a good choice for the presentation of our accomplishments.

Further Comments:

1) I suggest the authors use and apply not only FUS, they should include one or two more proteins, and prepare a compact comparative summary of the results obtained by the new sever.

 Authors’ reply: We agree with this suggestion; however, the time allowed for revision (5 days) and the current volume of the manuscript do not allow us to add another case study. We intend to familiarize the user with BIAPSS functionality empowering their LLPS-related analysis (SingleSEQ pipeline). So, we believe that the guidance and detailed analysis of the FUS case provides sufficient guide to the functionality and evidence of the comprehensiveness and usefulness of the BIAPSS service. The online documentation includes detailed instructions and tips for all applications. In addition, the MultiSEQ pipeline of the BIAPSS platform contains a comparative summary and some general insights, however, this is beyond the scope of this work.

2) They authors may highlight what the server requires as imput first, UNIProt ID ?

 Authors’ reply: The SingleSEQ pipeline of BIAPSS enables the analysis of a single protein sequence at once. In the main menu on the case summary page (https://biapss.chem.iastate.edu/single_seq.html), the user can directly select UniProt ID or filter a group of desired entries by querying a common protein name, gene identifier, and organism. If the number of matching cases is more than one, you can navigate between them using a horizontal slider, where each blue dot corresponds to a separate entry in the database. Querying the service is described in the manuscript for the FUS example, in Section 2.2, second paragraph between lines 168-175.

3) Any option to introduce a amino acid sequence ?

 Authors’ reply: In the current release, the BIAPSS server enables to browse and interactively explore the results for over 500 cases stored in an internal database. It is a large set of validated LLPS-related proteins covering the three primary databases (PhaSePro, PhaSepDB.v1, LLPSDB). To identify the UniProt ID of customized sequence, we suggest using BLAST, available online from the UniProt service (https://www.uniprot.org/blast/). BIAPSS's ability to analyze a sequence of any protein or artificially modified mutants is not currently available, and it is beyond the scope of this work. However, it is the main direction for further developing the BIAPSS's functionality.

4) And data obtained are accombined with a reliability factors, or score factors ?

For example Figure 1, left site, data shown in %, have also standard deviations ?

Authors’ reply: Many characteristics provided by applications in the SingleSEQ pipeline are qualitative and show a profile or pattern of the feature along the amino acid sequence. Examples include distributions of physicochemical characteristics, such as polarity, hydrophobicity, charge, residues forming hydrogen bonds, and pi-stacking. In these cases, the assignment is binary, and the numerical value is the percentage of residues in the sequence that meet the criterion. The second group of characteristics includes structural features predicted based on the amino acid sequences with top-ranked tools. Here, examples include secondary structure, solvent accessibility, the tendency to disorder, and low complexity regions. The visual representation is to assign each position along the amino acid sequence a discrete consensus value (e.g., a helical or extended, or coil for the secondary structure). The numerical value is the percentage of residues that meet the given criterion, e.g., % of helical. Figure 1 is a concept image, while in the interactive graphs, there is a label of what the given value refers to. The details are in the documentation. Furthermore, for those interested in in-depth analyses, the individual applications offer the user an on-the-fly exploration of the results from original tools, which typically provide fractional probabilities for each variant of a feature (e.g., p(helical), p(extended), p(coil)) for each position along the protein sequence. The corresponding clarification is added to the revised manuscript.

Reviewer 2 Report

This manuscript reports the development of an online visualization tool - BIAPSS - that has integrated available metadata on proteins annotated to be involved in liquid-liquid phase separation (LLPS). The tool would be widely applicable, easily accessible and would be attractive to researchers of broad interests in cell biology.

Some comments:

Lines 31-32 need to be rewritten for clarity.

The authors must clearly explain their criteria for inclusion/exclusion of a candidate LLPS protein - are proteins shown to be clearly driving phase separation included or any protein that is associated with LLPS included, etc.

There are several ways by which BIAPSS could be expanded into and for insights from BIAPSS to discover unknown LLPS proteins. It would be great for the authors to discuss such potential applications in a paragraph.

Author Response

Comments for the Authors

This manuscript reports the development of an online visualization tool - BIAPSS - that has integrated available metadata on proteins annotated to be involved in liquid-liquid phase separation (LLPS). The tool would be widely applicable, easily accessible and would be attractive to researchers of broad interests in cell biology.

Authors’ reply: Thank you for this comment.

Some comments:

1) Lines 31-32 need to be rewritten for clarity.

 Authors’ reply: The mentioned sentence has been removed from the manuscript for clarity. We believe the context of the paragraph remains consistent without it.

2) The authors must clearly explain their criteria for inclusion/exclusion of a candidate LLPS protein - are proteins shown to be clearly driving phase separation included or any protein that is associated with LLPS included, etc.

Authors’ reply: BIAPSS analyses were applied for over 500 proteins collected as a joint, non-redundant, and high-quality superset of proteins derived from several primary databases of curated LLPS deposits, including PhaSePro (proteins driving LLPS in living cells); PhaSepDB (LLPS-related proteins, manually curated); LLPSDB (proteins undergoing liquid-liquid phase separation in vitro with experimental validation, manually curated). Details are provided in the 3rd paragraph of section 2.1 Introduction of BIAPSS. Further, the summary page of the SingleSEQ analysis (https://biapss.chem.iastate.edu/single_seq.html) shows in which primary databases the sequence is present. That allows the user to assess the reliability more confidently whether the protein is driving LLPS or only associated with LLPS. Under different conditions or different compositions of biomolecules, both scenarios can be true, especially if the sequence is strongly multivalent.

3) There are several ways by which BIAPSS could be expanded into and for insights from BIAPSS to discover unknown LLPS proteins. It would be great for the authors to discuss such potential applications in a paragraph.

Authors’ reply: That is a great point. Thank you for this suggestion. We have added an appropriate paragraph in the Conclusions section.

When the current version of BIAPSS enables convenient and insightful analysis of large non-redundant and high-quality LLPS protein supersets, there are several directions through which our platform could expand to discover unknown LLPS proteins. That includes analyzing physicochemical and structural properties for customized protein sequences and introducing several LLPS indicators trained with machine learning to reveal coupled effects. The new functionality will find applications for flexible sequence redesign to introduce or modulate phase separation. Among many beneficial uses, it could tailor the properties of modern biomaterials or open-up new directions in the development of medical therapies.